# Cryopreservation of Woody Crops: The Avocado Case

**DOI:** 10.3390/plants10050934

**Published:** 2021-05-07

**Authors:** Chris O’Brien, Jayeni Hiti-Bandaralage, Raquel Folgado, Alice Hayward, Sean Lahmeyer, Jim Folsom, Neena Mitter

**Affiliations:** 1Centre for Horticultural Science, Queensland Alliance for Agriculture and Food Innovation, The University of Queensland, St Lucia, QLD 4072, Australia; j.hitibandalarage@uq.edu.au (J.H.-B.); a.hayward@uq.edu.au (A.H.); n.mitter@uq.edu.au (N.M.); 2The Huntington Library, Art Museum, and Botanical Gardens, 1151 Oxford Road, San Marino, CA 91108, USA; rfolgado@huntington.org (R.F.); slahmeyer@huntington.org (S.L.); jfolsom@huntington.org (J.F.)

**Keywords:** vitrification, ex situ conservation, long-term conservation, embryogenic, shoot tips, plant biodiversity

## Abstract

Recent development and implementation of crop cryopreservation protocols has increased the capacity to maintain recalcitrant seeded germplasm collections via cryopreserved in vitro material. To preserve the greatest possible plant genetic resources globally for future food security and breeding programs, it is essential to integrate in situ and ex situ conservation methods into a cohesive conservation plan. In vitro storage using tissue culture and cryopreservation techniques offers promising complementary tools that can be used to promote this approach. These techniques can be employed for crops difficult or impossible to maintain in seed banks for long-term conservation. This includes woody perennial plants, recalcitrant seed crops or crops with no seeds at all and vegetatively or clonally propagated crops where seeds are not true-to-type. Many of the world’s most important crops for food, nutrition and livelihoods, are vegetatively propagated or have recalcitrant seeds. This review will look at ex situ conservation, namely field repositories and in vitro storage for some of these economically important crops, focusing on conservation strategies for avocado. To date, cultivar-specific multiplication protocols have been established for maintaining multiple avocado cultivars in tissue culture. Cryopreservation of avocado somatic embryos and somatic embryogenesis have been successful. In addition, a shoot-tip cryopreservation protocol has been developed for cryo-storage and regeneration of true-to-type clonal avocado plants.

## 1. Introduction

Globally plants are recognized as a vital component of biodiverse ecosystems, the carbon cycle, food production and the bioeconomy. An estimated 7000 species of plants provide food, fiber, fuel, shelter and medicine [1]. Plant genetic diversity is the foundation of crop improvement [2] and a primary target of conservation efforts. The two major approaches to conserve plant genetic resources are ex situ and in situ conservation [3]. In situ conservation involves the designation, management and monitoring of target taxa where they are encountered [4]. It protects an endangered plant species in its natural habitat. In situ techniques are described as protected areas, e.g., genetic reserve, on-farm and home garden conservation. Ex situ conservation involves the sampling, transfer and storage of target taxa from the collecting area [4]. Ex situ techniques include seed, in vitro (tissue culture and cryopreservation), DNA and pollen storage; field gene banks and botanic garden conservation. In vitro storage using tissue culture and cryopreservation techniques can deliver valuable tools to achieve a positive conservation outcome for genetic resources.

The majority of conservation programs focus on seed storage [5]. Many of the world’s major food plants produce orthodox seeds which undergo maturation drying and are tolerant to extensive desiccation and can be stored dry at low temperature [6]. Seed storage under dry and cool conditions is the most widely adopted method for long-term ex situ conservation at relatively low costs [7]. About 45% of the accessions stored as seeds are cereals, followed by food legumes [15%], forages [9%] and vegetables [7%] [8]. However, seeds of many woody perennial plants are recalcitrant, e.g., *Juglans* spp. (walnut) [9], *Hevea brasiliensis* (rubber tree) and *Artocarpus heterophyllus* (jackfruit) [10]. Thus, they are difficult to maintain in seed banks. Additionally, seed-based conservation efforts miss clonal lineages that form the foundation of woody perennial agriculture [9]. Crops such as *Persea americana* Mill. (avocado), have recalcitrant seeds that are shed at relatively high moisture content, thus cannot undergo drying to facilitate long-term storage [6,11]. In addition, species that are seedless, e.g., *Musa* spp. (edible banana); or crops vegetatively propagated as their seeds are not true-to-type, e.g., *Manihot esculenta* (cassava), *Malus domestica* Borkh (Apple). and *Citrus* spp. (citrus)*;* are not storable through seeds. Field, in vitro and cryopreserved collections provide an alternative [7].

Field gene banks maintain living collections [12]. They are advantageous as physiological attributes and characteristics of the accessions such as plant habit, yield, tree height and disease resistance can be evaluated periodically [13]; however, there are several limitations posed; high maintenance cost, intensive labor and land requirements, pressure of natural calamities, risk of biotic and abiotic stresses as well as funding sources and economic decisions limiting the level of accession replication to maintain genetic diversity.

Tissue culture maintains plant material collections employing growth retardants [14], reduced light [15] or reduced temperature [16] to achieve slow growth, normally in sterile conditions. Plant germplasm storage via these methods has been increased with more in vitro protocols being developed for a vast number of plant species [17,18,19]. These approaches are used for large-scale micropropagation, reproduction purposes including embryo rescue, ploidy manipulations, protoplast fusions and somatic embryogenesis and are appropriate tools for short- and mid-term storage of plant genetic resources [7]. These methods allow for physical evaluation of material, rapid multiplication and plant establishment when needed, still, very costly to maintain due to space, consumables and labor inputs [20].

Plant cryopreservation (storage at −196 ± 1 °C) is a technique whereby plant tissues are preserved at ultra-low temperatures without losing viability [21]. It is the most relevant technology that provides safe long-term conservation of biological material as it maintains ex vivo biological function, does not induce genetic alterations [22] and provides long-term stable storage. Thus, it serves as an ultimate back-up of plant accessions for long-term storage, and material is generally not withdrawn from cryotanks unless it is necessary to use for research such as genetic manipulations [23] or in vitro culture [24]. A wide range of plant tissue can be cryopreserved, e.g., pollen, seeds, shoot tips, dormant buds, cell suspensions, embryonic cultures, somatic and zygotic embryos and callus tissue [25,26]. Recent uses of cryopreservation including cryotherapy to eradicate pathogens, such as phytoplasmas, viruses and bacteria in plants [27,28] is gaining a lot of attention [23]. Samples are normally given a short exposure to LN and surviving cells are regenerated from meristematic tissue which is pathogen free [28]. Cryotherapy has been used successfully in eradicating virus infections in several species with economic importance, such as *Prunus* spp. (plum), *Musa* spp. (banana), *Vitis vinifera* (grape), *Fragaria ananassa* (strawberry), *Solanum tuberosum* (potato), *Rubus idaeus* (raspberry) and *Allium sativum* (garlic) [28]. This review will look at conservation approaches for woody plants, focusing on avocado as a case study.

## 2. Field Repositories of Woody Crops

Field based germplasm conservation maintains living plants and serves as a source of plant genetic variation. Plants represented in these collections are current and historic cultivars, breeding material, landraces and sometimes wild relatives [9]. All of these are important to maintain for future development of new cultivars with superior growth characteristics or resistance to pest and diseases. Field repositories have the advantage that researchers can physically evaluate and characterize the accessions for parameters such as yield, tree height and disease resistance [29,30]. Table 1 summarizes some examples of woody crops that are held as field repositories. However, the field repositories require an adequate area of land and continuous maintenance as well as on-going funding. They are also vulnerable to loss from natural disasters and damage caused by pests and diseases. This makes it important to potentiate field germplasm conservation with other methods which address some of these concerns.

## 3. In Vitro Conservation

Different in vitro storage methods are employed depending on the storage duration required [17,35], i.e., in vitro culture for short- and medium-term storage and cryopreservation for long-term storage. Many reviews have been carried out to determine success [35,36,37,38] and standards established for managing field and in vitro germplasm gene banks [39,40]. These standards ensure effective, safe and efficient conservation of genetic resources. Due to the success of in vitro conservation techniques, many in vitro gene banks have been established nationally and internationally [41,42] (Table 2).

## 4. Plant Cryopreservation of Somatic Embryos and Shoot Tips

Cryopreservation of plants covers the entire plant kingdom from herbs and vines to shrubs and trees. The growth may be annual, biennial or perennial and the climate arctic; temperate, sub-tropical or tropical. A range of responses can occur within these groups and they are not always useful groupings for evaluating cryopreservation strategies [46]. The choice of material used, depends on the conservation goal, e.g., seeds and embryos capture species diversity; whereas shoot tips and dormant buds capture specific genotypes [47]. The most commonly used material to cryopreserve is apical meristems. They are at less risk of genetic variations due to their organized structure and are made up of small unvacuolated cells generally having a small vascular system [48]. In species that are recalcitrant and maintained in living field repositories, long-term cryopreservation storage of shoot tips can offer an alternative back-up as compared to seed storage which is only short-term [24].

Cryopreservation has several steps: (1) initial excision of the germplasm; (2) desiccation or pre-culture on osmotic media to reduce water content; (3) cryoprotection through exposure to cryoprotective agents; (4) cryopreservation in LN; (5) re-warming; and (6) unloading of cryoprotective agents and recovery of germplasm after cryopreservation [49]. The most critical step of cryopreservation is avoiding the intracellular and extracellular water that can lead to damage of cells during freezing [21]. Crystal formation, without extreme reduction of cellular water, can only be prevented though ‘vitrification’ i.e., the physical process of transition of an aqueous solution into an amorphous and glassy state (non-crystalline state) [50].

### 4.1. Methods to Reduce Water Content

Concentrated intracellular solute is a pre-requisite for successful cryopreservation and can be achieved with the following methods (Table 3) either individually or in combination [50,51,52,53].

Cryopreservation protocols using vitrification solutions typically involve a two-step cryoprotection process: (1) loading sometimes called osmoprotection is achieved by incubation in loading solution; and (2) dehydration using vitrification solution [52]. Loading solutions are commonly used to improve permeation of the cryoprotectant through cell membrane, it also induces tolerance to dehydration, which will be imposed by vitrification solutions. A common loading solution used is 2 M glycerol + 0.4 M sucrose [52]. Vitrification solutions contain chemicals that are high in concentration, e.g., ethylene glycol, glycerol and DMSO which have been reported as toxic to plant tissue [54]. It is therefore important to establish minimum exposure time to vitrification solutions in order to dehydrate tissue sufficiently to undergo cryopreservation and avoid damage effects to plant tissue [55,56].

Application of cryoprotectants is the most widely used method in cryopreservation protocols. Cryoprotectants that are penetrating in nature are able to reduce cell water at temperatures sufficiently to minimize the damaging effect of the concentrated solutes on the cells [57]. Whereas non-penetrating cryoprotectants osmotically “squeeze” water from the cells during the initial phases of freezing at temperatures between −10 and −20 °C [57]. Many authors have developed mixtures of cryoprotectants (Table 4) since the discovery of their benefits in protecting cells during the cryogenic process [54,58,59,60,61]. The most commonly used cryoprotectants for plant cells are PVS2 [59] and PVS3 [58].

### 4.2. Cryopreservation Methods

Presently there is no one method of cryopreservation that can be applied to a diverse range of plant species. Many cryopreservation methods (Table 5) have been developed for shoot tips and somatic embryos depending on the plant species used [17]; namely, vitrification, droplet-vitrification, encapsulation-vitrification, encapsulation-dehydration, dehydration, pre-growth, pre-growth-dehydration and D-cryoplate and V-cryoplate, a modification of the encapsulation-vitrification and droplet-vitrification [52,67,68].

#### 4.2.1. Vitrification

Vitrification can include the pre-culture of samples on medium supplemented with sucrose, then treated with a loading solution normally high in sucrose molarity [52] (e.g., a mixture of sucrose and glycerol), dehydration with a vitrification solution such as PVS2 or PVS3, rapid cooling, rewarming, and plant recovery by removing cryoprotectants [78].

#### 4.2.2. Droplet-Vitrification

Droplet-vitrification is a modification of vitrification [79]; treating explants with loading (usually 2 M glycerol and 0.4 M sucrose) and vitrification solutions; cooling them ultra-rapidly in a droplet of vitrification solution either PVS2 or PVS3 placed on an alfoil strip [49] with a droplet of cryoprotectant added before immersion in LN. The alfoil strip helps with the ultra-rapid cooling (about 4000–5000 °C min^−1^) and re-warming (3000–4500 °C min^−1^) of samples due to the good conductivity of thermal current of aluminum [80]. The removal of the cryoprotectant is achieved during re-warming stage by using an unloading solution usually with high level of sucrose 1.2 M, then transferred to recovery and regeneration media [25,55]. Droplet vitrification combines the use of highly concentrated vitrification solutions with ultra-fast cooling and re-warming rates [81] shown to be critical for survival [82]. For high success in survival and recovery of shoot tips after LN it is vital that samples are sufficiently dehydrated by the vitrification solution in order to vitrify while rapidly cooling in LN [83].

#### 4.2.3. Encapsulation-Vitrification and Encapsulation-Dehydration

Encapsulation-vitrification and encapsulation-dehydration have been successfully applied to cryopreserve shoot tips of woody species of crops, such as, *Malus* (apple) [84,85], *Pyrus* (pear), *Morus* (mulberry) [84], *Vitis* (grape) [86] and *Poncirus trifoliata* × *Citrus sinensis* (Chinese bitter orange) [87,88]. Dissected shoot tips or somatic embryos are suspended in a solution of sodium alginate. Beads (4–5 mm in size) are then formed using a truncated pipette tip and pipetted into a solution of CaCl_2_ where they are allowed to set for 30 min [52]. For encapsulation-vitrification, once beads are formed with explant inside, they are then dehydrated in PVS solutions such as PVS2 or PVS3 prior to immersion in LN. Although encapsulation is time-consuming, it eases manipulation due to alginate beads being relatively large in size [52]. For the encapsulation-dehydration technique instead of dehydration with PVS solutions beads are dehydrated in a laminar flow hood or under silica gel before immersion in LN [52].

#### 4.2.4. Dehydration

Of all the methods explained, dehydration is the simplest, as it involves just the dehydration of explants followed by direct immersion in LN. Embryonic axes or zygotic embryos extracted from seeds are mainly used. Desiccation is usually achieved by the air current of a laminar airflow cabinet or over silica gel. Dehydration using a vitrification solution removes intracellular water from cells and permits intracellular solution to undergo phase transition from liquid phase into an amorphous phase upon rapid cooling [52]. Cryoprotectant mixtures are commonly used as vitrification solution, such as PVS2 and PVS3.

#### 4.2.5. Pre-Growth and Pre-Growth-Dehydration

In pre-growth and pre-growth-dehydration, explants are first exposed and grown on media containing cryoprotectants, dehydrated by air under a laminar flow cabinet or with silica gel, and then frozen rapidly. Depending on the plant species optimal conditions can vary greatly.

#### 4.2.6. D-cryoplate and V-cryoplate

D-cryoplate and V-cryoplate use special aluminium cryoplates which have been developed (length 37 mm, width 7 mm and a thickness of 0.5 mm with 10 wells). An alginate solution containing 2% (*w*/*v*) sodium alginate in calcium-free MS basal medium with 0.4 M sucrose is poured over the cryo -plate. Samples are placed in wells and more sodium alginate solution is poured over the top to cover them. In V-cryoplate, dehydration is performed using the vitrification solution PVS2, while in D cryo-plate, dehydration is achieved using the air current of the laminar flow cabinet or silica gel [89]. After dehydration cryo-plates are immersed in LN. The main advantages of V-cryoplate and D-cryoplate is that handling of specimens is easy and quick because only the cryo-plates are manipulated [89].

## 5. The Avocado Case

### 5.1. Background

Avocado (*Persea americana* Mill.), a high-value fruit found in almost all tropical and sub-tropical regions of the world [90,91] belongs to the plant family Lauraceae [92], genus *Persea* [93]. Mexico is thought to be the center of origin of the species [94]. The genus *Persea* has about 400 to 450 species consisting of the currently often recognized genera *Alseodaphne* Nees, *Apollonias* Nees, *Dehaasia* Blume, *Machilus* Nees, *Nothaphoebe* Blume, *Persea* Mill. and *Phoebe* Nees. There are eight sub-species of *P. americana* including *P. americana* var. nubigena (Williams) Kopp, *P. americana* var. steyermarkii Allen, *P. americana* var. zenymyerii Schieber and Bergh, *P. americana* var. floccosa Mez, *P. americana* var. tolimanensis Zentmyer and Schieber, *P. americana* var. drymifolia Blake, *P. americana* var. guatemalensis Williams, *P. americana* var. americana Mill. [91,95]. Genetic diversity within the genus *Persea*, the sub-genera *Persea* and *Eriodaphne* and the species *P. americana* is large and is threatened by the progressive loss of tropical and sub-tropical forests [95]. This genetic diversity can serve as a resource in crop improvement [96,97,98] and plays an important role both ecologically and culturally.

The three recognized ecological races of *P. americana* [99]; are the Mexican race, *P. americana* var. drymifolia, adapted to the tropical highlands; the Guatemalan race, *P. americana* var. guatemalensis, adapted to medium elevations in the tropics; and the West Indian race, *P. americana* var. americana, adapted to the lowland humid tropics [100]. The ability of the three main races to withstand cold conditions varies; the West Indian race cannot tolerate temperatures below 15 °C, the Guatemalan race can tolerate cooler temperatures of −3 to −1 °C, and the Mexican race withstands temperatures as low as −7 °C exhibiting the highest cold tolerance [101,102,103]. They have distinctive characteristics; e.g., plant habit, leaf chemistry, peel texture, fruit color, disease and salinity tolerance [104]. The Guatemalan and Mexican races and their hybrids are very important for conservation and future breeding programs [97]. Cultivars classified as pure Guatemalan and Mexican races and Mexican × Guatemalan hybrids have been shown to have more diversity than those of pure West Indian race and Guatemalan × West Indian hybrid cultivars [97]. In Mexico and Central America, avocado trees grow under highly varied ecological conditions and natural selection over thousands of years has produced vast populations [97]. This serves as an essential source of varied attributes that are not among horticulturally available items [105].

The main avocado sold throughout the world, ‘Hass’, is a medium sized pear-shaped fruit with dark purplish black leathery skin [106]. Its commercial value is due to its superior taste, size, shelf-life, high growing yield, and in some areas, year-round harvesting [107]. The precise breeding history of ‘Hass ’ is unknown however, it is reported to be 61% Mexican and 39% Guatemalan [108]. This finding is supported by a study that analyzed the complete genome sequences of a ‘Hass’ individual and a representative of the highland Mexican landrace, *Persea americana* var. drymifolia; as well as genome sequencing data for other Mexican individuals, Guatemalan and West Indian accessions [108]. Analyses of admixture and introgression highlighted the hybrid origin of ‘Hass’, pointed to its Mexican and Guatemalan progenitor races and showed ‘Hass’ contained Guatemalan introgression in approximately one-third of its genome [108]. In Australia, ‘Hass’, represents 80% of total production [109] with 2019/20 producing 87,546 tonnes of avocados, an increase of 2% more than the previous season’s 85,546 tonnes [109]. This increased consumer demand is due to its popularity as a healthy food; often referred to as a superfood due to its beneficial nutrients, vitamins, minerals, fiber and healthy fats [110,111]. Consumer market value of Australian fruit sold domestically was worth ~$845 m in 2019/20 [109].

Due to the vast range of climates and conditions in our eight major avocado growing regions, avocados are produced all year round [109]. Avocado trees propagated by seed, take approximately 4–6 years to bear fruit, in some cases they can take 10 years to come into bearing [111]. Avocado trees are partially able to self-pollinate. Their flowers behave in synchronous dichogamy, flowers are perfect, bearing both male and female parts, however the periods of maleness and femaleness are temporarily distinct to enhance the likelihood of outcrossing [112,113]. The resultant progeny is highly heterozygous in the desirable parent tree characteristics [114]. New cultivars are normally derived from chance seedlings or mutations due to the difficult nature of breeding programs, which are costly, time-consuming and under threat of abiotic and biotic stresses. Nevertheless, the avocado industry’s goal is to preserve superior cultivars for commercial production. Thus, to meet this goal, avocado is propagated clonally through grafting with breeding programs based on both scion and rootstock cultivars. The threat of Ambrosia beetle species and its symbiont fungus Laurel Wilt disease to the avocado field gene banks and commercial industry in Florida, California, and Israel is a glaring example of a biotic stress that could destroy the industry [115]. For scion cultivars the focus is on high yield [116], extending harvest season, regular bearing tendencies and disease resistance e.g., Anthracnose [117], Cercospora spot [118] and Verticillium wilt [117]. Rootstocks are often selected for dwarf size [119], salinity tolerance, adaptation to alkaline soil [119,120] and pest and disease resistance [120] such as *Phytophthora cinnamomi* Rands and *Rosellina necatrix* [121]. Clonal rootstocks are thought to be the only rootstocks for the future for achieving sustainable productivity gains [122,123,124]. These influence the total productivity of the plant in terms of yield and health. Rootstocks from Mexico, ‘Orizaba 3’, ‘Antigua’ and ‘Galvan’, show a universal adaptation to multiple soil stress problems. The last two, also, have tolerance to *P. cinnamomi* [96]. Many breeding programs have concentrated on the development of new rootstocks such as ‘Dusa’, ‘Bounty’ and ‘Velvick’ [125] to help the industry overcome these threats [126]. ‘Dusa’s popularity has increased significantly since the mid-2000s. It is a common standard against which other *P. cinnamomi* tolerant rootstocks are compared in international breeding programs. It has been reported to bear fruit even under heavy *P. cinnamomi* disease pressure and has higher yields than many other rootstocks [127]. ‘Bounty’ is often selected for its *P. cinnamomi* tolerance and ability to survive in wet soils [127].

### 5.2. Avocado Conservation

#### 5.2.1. Global Germplasm Repositories

Field living germplasm collections (Table 6) and (Figure 1), are currently the most used conservation method, but funding and threats from natural calamities; pest and diseases are a problem.

#### 5.2.2. Cryopreservation of Avocado Somatic Embryos

To preserve global avocado diversity; development of improved technologies for avocado conservation, breeding/improvement and propagation is essential. In vitro somatic embryogenesis has direct importance to these objectives [139,140]. Somatic embryogenesis is the process by which somatic cells give rise to totipotent embryogenic cells capable of becoming complete plants [141]. Somatic embryogenesis can be a robust tool to regenerate genetically clonal plants from single cells chosen from selected plant material, or genetically engineered cells [142]. Somatic embryogenic cultures are generally highly heterogeneous since they consist of embryos at different developmental stages [143]. Though heterozygous in nature when regenerated using zygotic embryos as explants, cryopreservation of avocado somatic embryos offers an attractive pathway to conserve avocado germplasm. Recovery of plantlets from somatic embryos and clonal multiplication in vitro is an essential step for commercial application of this technology to crop improvement [144].

Somatic embryogenesis in avocado was first achieved using immature zygotic embryos of cv ‘Hass’ [145]. Studies have reported that the embryogenic capacity of avocado was highly genotype dependent [146]. To improve somatic embryogenesis previous studies have shown that several factors are vital for success, (1) composition of media, (2) hormone type and concentration, (3) type and concentration of gelling agent and (4) light intensity [147]. Morphogenic competence of somatic embryos has been reported to be lost 3–4 months after induction depending on the genotype [145,148]. In addition, the main factor limiting conversion of somatic embryos into plantlets is incomplete maturation [149]. Studies have found that there are two types of regeneration that occur after maturation; unipolar (only shoot apex or root) and bipolar (both shoot apex and root). Shoots regenerated from unipolar embryos can either be rooted or rescued using in vitro micrografting [150]. Studies have shown that the percentage of high-quality bipolar embryos from avocado somatic embryos was extremely low at 2–3% and was genotype dependent [145,150,151]. This low rate of somatic embryo conversion is currently the main bottleneck in avocado regeneration via somatic embryogenesis [144]. A study described an in vitro induction and multiplication system for somatic embryos of avocado, across four cultivars, which remained healthy and viable for 11 months, on a medium used for mango somatic embryogenesis [139]. Furthermore for one of the cultivars, cultivar ‘Reed’, a two-step regeneration system was developed that resulted in 43.3% bipolar regeneration [139].

Cryopreservation of avocado somatic embryos has been successful for various cultivars (Table 7). The effect of cryogenic storage on five avocado cultivars (‘Booth 7’, ‘Hass’, ‘Suardia’, ‘Fuerte’ and ‘T362’) using two cryopreservation protocols (controlled-rate freezing and vitrification) was investigated [152]. In terms of controlled-rate freezing, three out of five embryogenic cultivars were successfully cryopreserved with a recovery of 53 to 80%. Using vitrification, cultivar ‘Suardia’ showed 62% recovery whereas ‘Fuerte’ had only a 5% recovery. When the droplet-vitrification technique was used, two ‘Duke-7’ embryogenic cell lines showed viability ranging from 78 to 100% [153]. Protocols employed in both studies cannot be applied in general to multiple cultivars and optimization of loading sucrose concentrations and plant vitrification solution 2 (PVS2), temperature and times need more intensive research.

#### 5.2.3. Shoot-Tip Cryopreservation of Avocado

Cryopreservation is a secure and cost-effective method for long-term storage of avocado. It provides a high degree of genetic stability in maintaining avocado collections for the long-term compared to other conservation methods. Shoot-tip cryopreservation conserves ‘true-to-type’ avocado plant tissue. It is ideal for preserving a core selection of avocado genotypes, for example, with superior characteristics, disease and pest resistance, rarity, drought and salinity tolerance. In one study, it was shown that axillary buds of Mexican and Guatemalan races were viable through fluorescein diacetate staining after dehydration with sterile air and being treated with cryopreservation solutions; however, shoot regeneration was not achieved with the cryopreserved material [154]. Another study, showed that dehydration at 60 min with sterile air and 30 min in PVS4 at 0 °C produced normal plant development and 100% survival was obtained after 30, 45 and 60 days [155].

#### 5.2.4. Critical Factors Identified for Successful Cryopreservation of Avocado Shoot-Tips

Although still cultivar-dependent, in vitro protocols have been established for multiple cultivars of avocado [111] advancing cryopreservation of avocado. Droplet vitrification can be considered as a “generic” cryopreservation protocol for hydrated tissues, such as in vitro cultures [49,156]. Vitrification-based procedures offer practical advantages in comparison to classical freezing techniques and are more appropriate for complex organs e.g., avocado shoot tips, which contain a variety of cell types, each with unique requirements under conditions of freeze-induced dehydration [157]. A problem associated with cryopreservation is formation of lethal ice crystals. To overcome this vitrification makes use of the physical phase called ‘vitrification’, i.e., solidification of a liquid forming an amorphous ‘or glassy’ structure [7] to avoid ice crystal formation of a watery solution. Glass is viscous and stops all chemical reactions that require molecular diffusion, which leads to dormancy and stability over time [158]. Samples can be vitrified and rapidly supercooled at low temperatures and form in a solid metastable glass with crystallization [66]. For procedures that involve vitrification, cell dehydration occurs using a concentrated cryoprotective media and/or air desiccation and is performed first before rapid freezing in LN [157]. It is important that cells are not damaged or injured during the vitrification process and are vitrified enough to sustain immersion in LN [24]. As a result, all factors that affect intracellular ice formation are avoided [157].

Oxidative stress is a common and often severe problem in plant tissue [159,160] of most woody plant species, such as avocado. Therefore, it is important to optimize regrowth conditions of extracted avocado shoot tips to prevent browning when developing an in vitro cryopreservation protocol. Browning of cell tissue takes place as the cytoplasm and vacuoles are mixed and phenolic compounds readily become oxidized by air, peroxidase or polyphenol oxidase. Oxidization of phenolic compounds inhibit enzyme activity and result in darkening of the culture medium and subsequent lethal browning of explants [161]. The antioxidant ascorbic acid (ASA) or vitamin C (ASA) occurs naturally in plants, in plant tissue and meristems [162]. It has many roles in a plant’s physiological processes but mainly in its defense against oxidative damage resulting from aerobic metabolism, photosynthesis, pollutants and other stresses caused by the environment [163]. Wounding of avocado tissue can lead to an increase in reactive oxygen species (ROS) within the shoot therefore affecting the viability. ROS are highly reactive molecules and have been shown to cause damage in cells. Many molecules are considered as ROS, some of which include oxygen-free radical species and reactive oxygen non-radical derivatives [48]. The most common ROS species found in plants are superoxide (O_2_^−^), hydroperoxyl (OOH), hydroxyl radical (OH) and singlet oxygen (O_2_) [48]. ASA has an important role in the detoxification of ROS species both enzymatically or non-enzymatically [164]. It can do this by scavenging a singlet oxygen, hydrogen peroxide, superoxide and hydroxyl radical [163].

It has been reported by several authors that the addition of antioxidants can help increase the viability of plants by suppressing browning which leads to shoot tip death [83,165,166,167,168,169]. By maintaining a higher antioxidant level protection improved post cryopreservation [166]. It has been reported that in *Actinidia* spp. (kiwifruit) the addition of ASA in regrowth media improved the survival after cryopreservation by reducing lipid peroxidation [83]. The addition of ASA to pre-culture media, loading solution, unloading solution and regrowth media significantly increased regrowth of shoot tips of *Rubus* spp. (raspberry) [168]. A recent study found treating *Persea americana* cv ‘Reed’ (avocado), with varying concentrations of different antioxidants (ASA, polyvinylpyrrolidone [PVP], citric acid and melatonin) reduced browning caused when extracting shoot tips. The type of antioxidant and concentration had an effect on viability, vigor and health of the shoots [170].

Avocado is highly susceptible to osmotic stresses imposed by cryoprotectants which are high in osmolarity. Cold sensitive species such as avocado are likely to be positively responsive to vitrification treatments during cryopreservation if optimizations are done carefully [171]. In order to improve on tolerance to cryoprotectants and increase permeation of the cryoprotectant through the cell membrane and induce tolerance to dehydration caused by vitrification solutions, a pre-step called ‘loading’ is used [52]. Loading is achieved by incubating tissues for 10−20 min in solutions composed of glycerol and sucrose [48]. This loading step is particularly useful for plant species, that are sensitive to direct exposure to cryoprotectants due to dehydration intolerance and osmotic stresses [48]. However, use of loading solution alone for avocado shoot tips is not adequate to induce tolerance to cryoprotectants, and other pre-treatments/pre-culture such as osmotic conditioning with sugars and cold acclimatization are necessary [172].

Pre-culturing shoot tips with a high sugar enriched media has been reported previously by several authors [173,174,175] to increase the viability post-cryopreservation by better pre-conditioning the shoot. Also, time of incubation in pre-culture solutions was critical to ensuring survival and high regrowth rates [55,176]. There have been attempts to use alternative sources of sugar in pre-culture media, such as, sorbitol or mannitol [177,178,179,180], glucose and fructose; all have shown no negative effects on post-cryopreservation survival [181]. However, most researchers prefer to use sucrose as the sugar source when adding to pre-culture media [181]. Sucrose has been found to be more beneficial in pre-culture as compared to sorbitol and mannitol as these two sugars were unable to support regrowth of olive somatic embryos [182]. However, when 0.2 M sorbitol was combined with 5% DMSO it was an effective cryoprotectant for embryogenic tissue of *Pinus roxburghii* Sarg. (chir pine) [183]. Sucrose is an excellent glass former and is able to stabilize membranes and proteins [184]. Sucrose stimulates the production of other elements such as proline, glycine betaine, glycerol and polyamines, which have colligative as well as non-colligative effects [185,186]. Of the above-mentioned sugars [187], glycerol [188], proline [189] and glycine betaine [190] have proved their cryoprotectant ability, whereas polyamines are known for their antioxidant properties. Therefore, these compounds play a vital role in protecting the cells during cryopreservation. It has also been shown that pre-culturing in high sucrose media enhances the acclimatization process to low temperature and stimulates osmotic dehydration [47].

Water availability and temperature are influenced by environmental variables and are major determinants of plant growth and development [191]. Most tropical and sub-tropical species have little to no freezing tolerance, however, temperate plant species have evolved some form of cold tolerance [191,192]. It has been shown in temperate plants that they have the genetic ability to increase cold tolerance significantly when exposed to environmental cues that signal the arrival of winter [193]. Many plants can increase their tolerance to the cold by exposure to lower temperatures, generally with temperatures below 10 °C [193]. This process is referred to as cold hardening or cold acclimatization (CA) and requires days to weeks for full development [50,193,194]. Several biochemical, physiological and metabolic functions are altered in plants by low temperature as well as gene expression [195]. Expression of cold induced genes include those that control the function of cell membranes to stabilize and protect themselves against freezing injury [196]. Freezing tolerance can be increased by 2–8 °C in spring annuals, 10–30 °C in winter annuals and 20–200 °C in tree species [193]. Cold acclimatization can help improve the regrowth rates of in vitro plants, improve regeneration rates [197]. Cold acclimatization has been used as an in vitro pre-treatment on donor plants before shoot tip extraction [198] in developing cryopreservation protocols in plants such as *Malus domestica* Borkh (apple), *Malus sieversii* (Ledeb.) (wild apple) and *Phoenix dactylifera* (date palm) [199,200]. Cold acclimatization with or without ABA significantly improved the survival of *Rubus* spp. [201]. Abscisic acid (ABA) pre-treatment alone could not increase the survival of plants grown under warm conditions after cryopreservation, but the survival tripled when cold acclimatization was combined with ABA pre-treatment [201]. High sucrose (0.3 M) or low temperature (10 °C) incubation treatments primed in vitro plants of cvs ‘Reed’ and ‘Velvick’ shoot tips to tolerate cryoprotectant (PVS2) treatments but was cultivar-specific [202].

## 6. Conclusions

Field living germplasm collections are currently the only conservation method for avocado, but funding and threats from natural calamities; pest and diseases are a problem. Cryopreservation is an invaluable tool that could be utilized in conjunction with field repositories to securely preserve this important horticultural crop. There have been significant improvements within the cryopreservation platform to preserve *Persea* spp. germplasm [202,203,204]. Studies have shown that cryopreservation of somatic embryos offers usefulness in conserving *Persea* germplasm biodiversity [144,152,153]. An important factor for somatic embryos is that regeneration can be achieved after exposure to LN to ensure that protocols can be effectively applied for conservation programs [176]. Cryopreservation of somatic embryos is valuable as it is readily retrievable for further biotechnology manipulations as well as storage of biotechnology products such as genetically transformed lines [23,205].

To date, although cultivar-dependent, in vitro multiplication protocols have been established for maintaining multiple avocado cultivars in tissue culture from mature glasshouse cuttings [111]. This can be used to supply new plants to avocado farmers, meeting a critical issue that is preventing the expansion of industry, the shortage of available avocado trees. Twenty thousand in vitro plants can be maintained in a 10 sqm tissue culture room saving on land, fertilizer, pesticides promoting an environmentally sustainable and efficient method of multiplication of avocado plants.

Development of the in vitro shoot-tip cryopreservation protocol was highly dependent on the availability of this reliable in vitro multiplication and regeneration protocol. For the first time studies [202,203,204] have shown that in vitro cryopreservation using droplet-vitrification for mature material of two avocado cultivars have been successful. Correctly treating avocado shoot tips with the ideal pre-treatment before LN is vital for a successful outcome [202]. It was identified that the use of 100 and 250 mg L^−1^ of ASA can effectively reduce browning of freshly extracted avocado shoot tips [170,202]. High sucrose and cold pre-treatments are effective in increasing survivability following cryoprotectant incubation of avocado shoot tips. While pre-treatments are effective for avocado, the type of pre-treatment needed and the degree of effectiveness was cultivar-specific [202]. This can be directly linked to the genetics of the two cultivars which display varying tolerance to cold and salinity in their natural growing environments; namely, cv ‘Velvick’ from West Indian race (no cold tolerance) and cv ‘Reed’ from Guatemalan race (moderate cold tolerance) [204]. The type of cryoprotectant and exposure time to the cryoprotectant was also essential in obtaining morphologically normal and vigorous plants [204]. Avocado shoots that survived LN grew into full plants ready for rooting after 24 weeks [204]. Cultivar ‘Reed’ shoots were successfully rooted [206] and after 8 weeks, plantlets were ready to be acclimatized in a University of Queensland glasshouse (Figure 2). These plants will be screened for growth parameters and yield in a field trial at Duranbah, Queensland. Shoot tips from cv ‘Velvick’ are currently in the rooting stage.

In vitro multiplication and in vitro cryopreservation protocols provide another set of tools that can be used to preserve global avocado diversity to improve conservation germplasm collections, breeding and propagation. Somatic embryogenesis, cryopreservation of somatic embryos and shoot tips, have the ability to be adapted to lead to the establishment of a global Cryo-Bank conserving avocado biodiversity and offering a source of disease-free genetic material. They provide useful tools for further optimization of the species and other woody plant species facing similar challenges in conservation. Shoot tip cryopreservation is ideal for preserving a core selection of avocado genotypes, for example, with superior characteristics, disease and pest resistance, rarity, drought and salinity tolerance. Shoot tip cryopreservation of avocado is a major breakthrough and this work can pave the way for storing a core collection of *Persea* spp. for true-to-type avocado shoot tip preservation.

## Figures and Tables

**Figure 1 plants-10-00934-f001:**
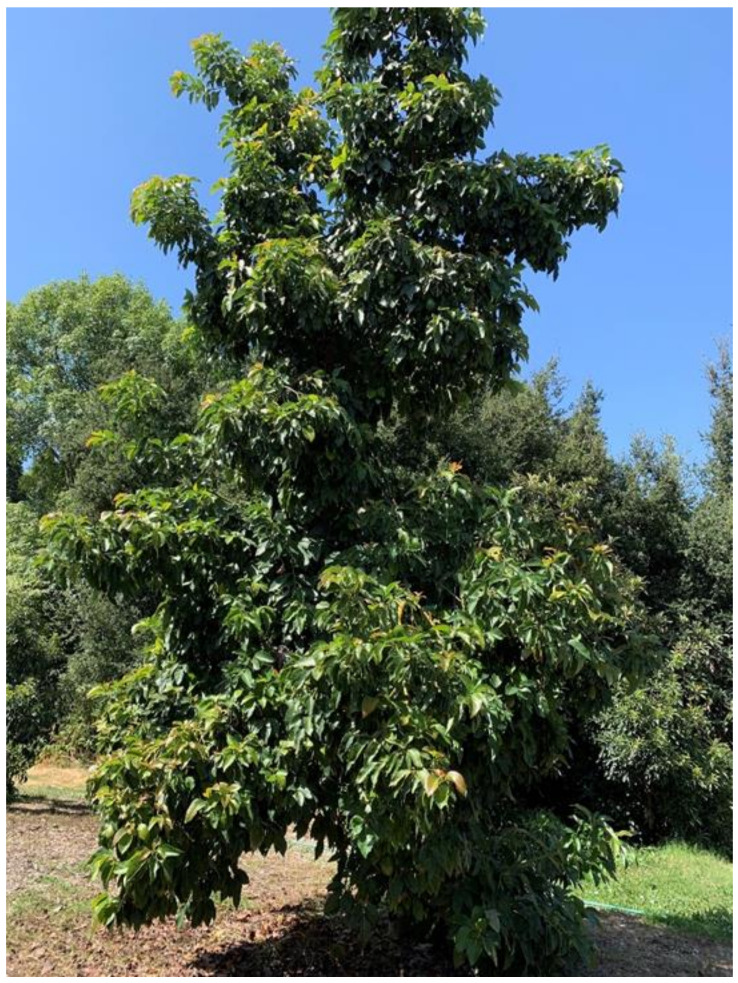
One of the 56 avocado accessions being maintained in The Huntington Botanical Gardens [in San Marino, California USA] living germplasm collection.

**Figure 2 plants-10-00934-f002:**
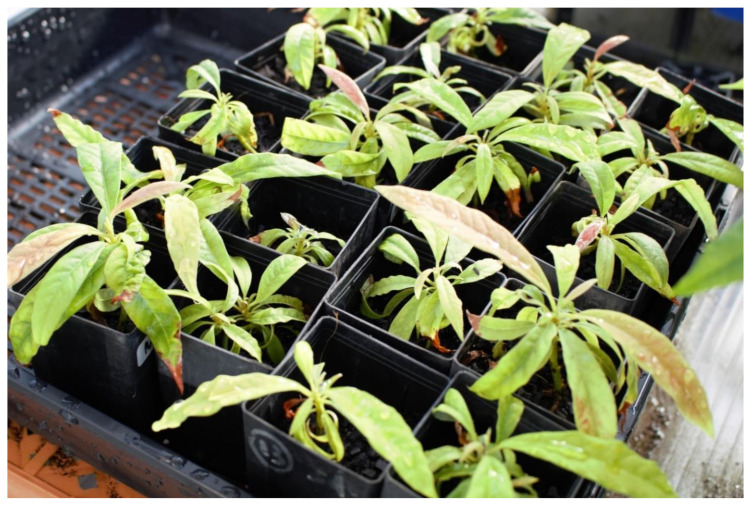
Shoot tips of cv ‘Reed’ treated with VSL and revived from LN growing in a glasshouse.

**Table 1 plants-10-00934-t001:** Some examples of field repositories maintaining living collections of economically important crops.

Country	Field Repositories	Genus/Species	Reference
USA	USDA—Geneva NY, Davis CA, Riverside CA	*Malus domestia* Borkh. (apple) *Vitis vinifera* L. (grape) *Actinidia deliciosa* (kiwifruit) *Diospyros* spp. (persimmon) *Ficus carica* L. (fig) *Juglans* spp. (walnut) *Olea europaea* L. (olive) *Pistacia vera* L. (pistachio) *Punica granatum* L. (pomegranate) *Citrus* spp. (citrus) *Prunus* spp. (plum)	[31]
USA	Tropical Botanical Garden	*Artocarpus altilis* (breadfruit)	[32]
Germany	German Fruit Gene bank	*Malus* spp. (apple) *Prunus avium* (cherry) *Prunus domestica* (plum) *Rubus* spp. (raspberry)	[12,33]
United Kingdom	National Fruit Collection	*Malus domestica* Borkh. (apple) *Prunus domestica* (plum) *Pyrus communis* L. (pear) *Prunus avium* (cherry)	[34]

**Table 2 plants-10-00934-t002:** Some examples of cryo-storage gene banks maintaining collections of economically important crops.

Country	Gene Bank	Genus/Species	Accessions Held	Reference
France	Institute of Research Development	*Coffea* spp. (coffee)	~500	[12]
Columbia	International Centre for Tropical Agriculture	*Manihot esculenta* (cassava)	5690	[43]
Japan	National Institute of Agrobiological Sciences	*Morus* spp. (mulberry) *Juncus effusus* (rush)	~1000 50	[12]
Japan	Shimane Agriculture Research Centre	*Wasabi japonica* M. (Japanese horseradish)	40	[12]
USA	National Clonal Germplasm Repository	*Malus* spp. (apple) *Pyrus* spp. (pear) *Rubus* spp. (raspberry) *Vitis* spp. (grape)	6073 131 57 1405	[44,45]
Belgium	Bioversity International Transit Centre	*Musa* spp. (banana)	1600	[7]

**Table 3 plants-10-00934-t003:** Methods to reduce water content.

Dehydration Method	Uses
Desiccation	(1) Air drying of explants in laminar flow hood or using flow of compressed air. (2) Dehydration of explants in a desiccator with silica gel.
Cryoprotectants	(1) Penetrating cryoprotectants, e.g., dimethyl sulfoxide (DMSO) and glycerol act by replacing intracellular water. (2) Non-penetrating cryoprotectants, e.g., sucrose, polyvinylpyrrolidone (PVP) and polyethylene glycol (PEG), display different osmotic potential inside and outside the cells.
Freeze-induced dehydration	Preferential freezing of extracellular water by slow cooling at a rate of 0.5–2 °C per min creates a hypotonic surrounding for the cell, resulting in outflow of cellular water.
Pre-conditioning of donor plant or explant	Including DMSO abscisic acid, sucrose, polyols or proline in the pre-culture medium or low temperature treatment to induce tolerance to dehydration and freezing.

**Table 4 plants-10-00934-t004:** Some examples of cryoprotectants used for plant tissue.

Cryoprotectant	Composition
PVS1	30% *w/v* glycerol, 15% *w/v* EG, 5% *w/v* sucrose, 15% *w/v* DMSO [61]
PVS2	30% *w/v* glycerol, 15% *w/v* DMSO, 15% *w/v* EG and 15% sucrose [59]
PVS3	50% *w/v* glycerol and 50% *w/v* sucrose [58]
PVS4	35% *w/v* glycerol, 20% *w/v* EG and 20.5% M sucrose [62]
VSL+	20% *w/v* glycerol, 10% *w/v* DMSO, 30% *w/v* EG, 15% sucrose and 10 mM CaCl_2_ [63]
VSL	20% *w/v* glycerol, 10% *w/v* DMSO, 30% *w/v* EG, 5% sucrose and 10 mM CaCl_2_ [63]
Steponkus	50% *w/v* EG, 15% sorbitol, 6.0% bovine serum albumin, 13.7% sucrose [64]
Towill	35% EG, 6.8% *w/v* DMSO, 10% PEG 8000 and 13.7% sucrose [65]
Fahy	20% DMSO, 20% formamide, 15% propylene glycol [66]

**Table 5 plants-10-00934-t005:** Some examples of cryopreservation methods, techniques and applications used.

Method	Technique	Application	Survival/Recovery	Reference
Vitrification	Pre-culture of cultures on basal medium supplemented with cryoprotectants, pre-treatment with loading solution, dehydration with PVS, rapid freezing rewarming.	Cocoa secondary somatic embryos	74.5% survival with 5- day pre-culture on 0.5 M sucrose followed by 60 min dehydration in PVS2 treatment for 1 h at 0 °C.	[69]
Droplet-vitrification	Resembles vitrification in all steps with only difference that materials are cryopreserved on foil strips in drops of vitirification solution.	*Hancornia* speciosa Gomes (rubber tree) shoot tips	43% regrowth with pre-culture on basal + proline (0.193 M) for 24 h in the dark at 25 °C and PVS2 15 min at 0 °C.	[70]
Encapsulation-vitrification	Sodium alginate beads are formed and explants are encapsulated in them and dehydrated in PVS before freezing.	*Olea europaea* (olive) somatic embryos *Parkia speciosa* Hassk. (stink bean) shoot tips	64% regrowth after 4 day pre-culture in sucrose; PVS2 treatment for 3 h treatment and rapid freezing. Pre-culture on MS + trehalose (5% *w*/*v*) for 3 days; PVS2 for 1 h at 0 °C.	[71] [72]
Encapsulation-dehydration	Sodium alginate-encapsulated cultures are dehydrated osmotically with high concentrations of sucrose for 1–7 days and/or desiccated in an air current before slow cooling to –80 °C and then immersed in LN.	*Olea europaea* (olive) somatic embryos *Prunus armeniaca* (apricot) shoots	40% regrowth following 4 days of sucrose pre-growth, desiccation and freezing. Recovered after treated with 0.5 M sucrose for 2 days followed by air dehydration for 2 h and frozen in LN.	[71] [73]
Dehydration	Samples are dehydrated by either air current, silica gels, or incubation with cryoprotectant, followed by rapid freezing or two-step freezing.	*Juglans nigra* (walnut) embryo axes	Dried in a laminar flow hood until 5–15% moisture content and 100% recovery after LN.	[74]
Pre-growth and pre-growth-dehydration	Samples are cultured on media containing cryoprotectants such as DMSO, dehydrated and then frozen slowly or rapidly.	*Garcinia mangostana* L. (mangosteen) shoot tips	50% MS + sucrose (0.6 M) + 5% DMSO for 2 days	[75]
V-cryoplate	Modification of encapsulation-vitrification and droplet-vitrification. Dehydration is performed using vitrification solution PVS2.	*Morus alba* (mulberry) shoot tips	87% regrowth, 13 lines pre-cultured at 25 °C for 1 day on MS medium containing 0.3 M sucrose. PVS2 solution for 30 min at 25 °C.	[76]
D-cryoplate	Modification of encapsulation-vitrification and droplet-vitrification. Dehydration is achieved using the air current of the laminar flow cabinet or silica gel.	*Diospyros kaki* (persimmon) shoot tips	Average 87% regrowth, 10 lines 1–3 months cold acclimatization, 3 °C pre-cultured on 0.3 M sucrose, 2 days at 25 °C, laminar flow 30 min at 25 °C.	[77]

**Table 6 plants-10-00934-t006:** Avocado germplasm maintained as field repositories throughout the world.

Country	Germplasm Repositories	No. of Accessions	References
USA	The Huntington San Marino CA	56 *Persea americana* accessions 4 wild *Persea* spp (6 accessions)	[128]
USA	Riverside University CA	~230 avocado scion accessions	[129]
		~15 wild *Persea* spp.	
		~246 avocado rootstock accessions	[129,130]
USA	National Genetic Resources Program, Miami, Florida	*P. americana* (167 accessions) and *P. schiedeana* (1 accession)	[44,131]
USA	The Sub-Tropical Horticulture Research Station, Miami, Florida	~400 avocado accessions	[132]
Mexico	National Research Institute of Forestry and Livestock in Guanajuato	500 accessions belonging to *P. americana*: Mexican and Guatemalan races. Related species: *P. schiedeana, P. cinerascens, P. floccosa, P nubigena*	[133]
Mexico	State of Mexico of the Fundación Salvador Sanchez Colin-CICTAMEX, S.C.	800 accessions of avocado and related species. Mexican, Guatemalan, West Indian races, *P. americana* var. costaricensis race materials.	[133]
Mexico	Coatepec Harinas and Temascaltepec; State of Mexico	Wild relatives: *Beilschmiedia anay, B. miersii, P. schiedeana, P. longipes, P. cinerascens, P. hintonni, P. floccosa, P. tolimanensis, P. steyermarkii, P. nubigena, P. lingue, P. donnell-smithii, P. parvifolia, P. chamissonis, Persea* spp.	[133]
Ghana	University of Ghana Forest and Horticultural Crops Research Centre	110 local land races and 5 varieties from South Africa (‘Hass’, ‘Fuerte’, ‘Ryan’, ‘Ettinger’ and ‘Nabal’	[134]
Israel	Volcanic Centre in Bet Dagan	194 trees, propagated from 148 accessions	[96]
Spain	The Experimental Station ‘La Mayora’ in Malaga	75 avocado accessions	[132,135]
Cuba	N/A	210 genotypes	[132]
Chile	N/A	4 botanical breeds of *P. americana*: var. drymifolia, var. guatemalensis, var. jacket and var. costaricencis	[132]
Australia	Maroochydore Research Station	46 avocado accessions	[136]
Nigeria		8 avocado accessions	[137]
Brazil	Brasilia, in the Federal District, depending on the Embrapa Research Institute	30 avocado accessions	[138]
Brazil	Conceicao do Almeida and Juazeiro collections, both in the Bahia State	22 avocado accessions	[138]
Brazil	Piracicaba, in the Sao Paulo State	33 avocado accessions	[138]
Brazil	Jaboticabal, in the Sao Paulo State	7 avocado accessions	[138]

**Table 7 plants-10-00934-t007:** Summary of successfully applied cryopreservation techniques to avocado somatic embryos. * Recovery is defined as any somatic embryo clump which was proliferating into new callus clumps.

Cryopreservation Technique	Cultivars	* Recovery Percentages
Vitrification	‘Suardia’	62%
	‘Fuerte’	5% [152]
	‘A10’	91%
	‘Reed’	73%
	‘Velvick’	86%
	‘Duke 7’	80% [144]
Slow freezing	‘Suardia’	60–80%
	‘T362’	4–53%
	‘Fuerte’	73–75% [152]
Droplet vitrification	‘A10’	100%
	‘Reed’	85%
	‘Velvick’	93% [144]
	Two lines of ‘Duke 7’	78–100% [153]

## Data Availability

Not applicable.

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
