# Peer review of "Cryopreservation of Woody Crops: The Avocado Case"

_plants, 2021, doi:10.3390/plants10050934_

Round 1
Reviewer 1 Report
Major changes introduced by the authors made great improvement on the presentation style of the manuscript. This is now a comprehensive, logically structured review which reads well and provides important information in the context of tree species conservation with the emphasis on avocado. My recommendation is to accept the paper with a few minor edits as below
- The combination “in vitro cryopreservation” (Lines 24, 394 and 400) may confuse the readers about the subject: in vitro propagation or cryopreservation. Consider removing “in vitro” from these sentences.
- The journal system detected errors with some references (Lines 117, 150 and further in the text). I could not find any issue with the references cited but please double-check.
- Table 3. “Cryoprotectants”. The statement “.. DMSO and glycerol have rapid penetrating power” is formally incorrect. There is a great difference between DMSO and Glycerol ability to penetrate plant cells and organs. For example, 90% of maximum DMSO concentration in Allium shoot tips can be achieved within 5 min of incubation with PVS2, while glycerol will take ca 30 min to penetrate. Consider removing the part of a sentence about the penetrating rate.
- Same location. “non-penetrating cryoprotectants .. dehydrate cells by extreme osmotic stress”. It is better to use “different osmotic potential inside and outside the cells” instead of “osmotic stress”. “Stress” is an environmental factor (or combination of factors). In this sentence you are talking about the physical process of dehydration, so better to use physical terms to describe it.
- Same location. Consider removing no. 3 (sentence about PVS2 and PVS3) from the table. You have mentioned these VS in the text, no need to repeat.
- Line 154. “loading” is often called “osmoprotection” in the literature. Consider changing or using both terms, e.g. “loading (sometimes called “osmoprotection”)”
- Please change “thawing” to “rewarming” throughout the text. “Thawing” is the physical term indicating the process of phase transition from ice to liquid which is what most people try to avoid during cryopreservation. It is generally recommended to use “rewarming” in papers describing cryopreservation through vitrification.
- Line 178-179. The generally accepted names of the methods are “D-cryoplate” and “V-cryoplate” methods.
- Table 5. Droplet-vitrification. In the sentence “Resembles vitrification…” please add “in drops of vitrification solution” at the end. This is an important detail to note.
- Line 303. “abiotic stress” is most likely a typo. The sentence is about biotic stress (pathogens).
- Table 6. Institutions are missing for Cuba and Chile. No information?
- Fig. 1, legend. “One of the many avocado accessions..”. Better to put the number: “One of 60 (correct?) avocado accessions..”
- Line 399-400. “Without a reproducible in vitro system, the in vitro cryopreservation protocols will never be successful”. This statement is disputable. What about using cryopreservation without in vitro? USDA developed the system when cryopreserved Citrus buds are grafted directly to greenhouse-grown rootstocks without recovery in vitro. Could this work for avocado? Consider removing this sentence.
- Lines 541-545. From my perspective, the information given here would be better placed before conclusion, when describing cryopreservation methods for avocado. Same for Fig. 2.
- Figure 1 is not mentioned in the text. Please check.
- Please fill in required sections (Authors contributions, Funding, Institutional review board statement, Data availability statement and Conflict of interests).
Author Response
The authors would like to thank the reviewer for their suggestions in improving the manuscript.
- The combination “in vitro cryopreservation” (Lines 24, 394 and 400) may confuse the readers about the subject: in vitro propagation or cryopreservation. Consider removing “in vitro” from these sentences. The term in vitro was removed from the 3 sentences.
- The journal system detected errors with some references (Lines 117, 150 and further in the text). I could not find any issue with the references cited but please double-check. The table references were added in text.
- Table 3. “Cryoprotectants”. The statement “.. DMSO and glycerol have rapid penetrating power” is formally incorrect. There is a great difference between DMSO and Glycerol ability to penetrate plant cells and organs. For example, 90% of maximum DMSO concentration in Allium shoot tips can be achieved within 5 min of incubation with PVS2, while glycerol will take ca 30 min to penetrate. Consider removing the part of a sentence about the penetrating rate. The part of sentence which referred to penetration rate was removed.
- Same location. “non-penetrating cryoprotectants .. dehydrate cells by extreme osmotic stress”. It is better to use “different osmotic potential inside and outside the cells” instead of “osmotic stress”. “Stress” is an environmental factor (or combination of factors). In this sentence you are talking about the physical process of dehydration, so better to use physical terms to describe it. The term osmotic stress was removed and replaced with different osmotic potential inside and outside the cells.
- Same location. Consider removing no. 3 (sentence about PVS2 and PVS3) from the table. You have mentioned these VS in the text, no need to repeat. Section 3 of this table was removed from the manuscript.
- Line 154. “loading” is often called “osmoprotection” in the literature. Consider changing or using both terms, e.g. “loading (sometimes called “osmoprotection”)” The term osmoprotection was added.
- Please change “thawing” to “rewarming” throughout the text. “Thawing” is the physical term indicating the process of phase transition from ice to liquid which is what most people try to avoid during cryopreservation. It is generally recommended to use “rewarming” in papers describing cryopreservation through vitrification. The term thawing was replaced with rewarming throughout the manuscript.
- Line 178-179. The generally accepted names of the methods are “D-cryoplate” and “V-cryoplate” methods. The names of the two techniques were changed as reviewer suggested.
- Table 5. Droplet-vitrification. In the sentence “Resembles vitrification…” please add “in drops of vitrification solution” at the end. This is an important detail to note. "in drops of vitrification solution" was added to the end of the phrase.
- Line 303. “abiotic stress” is most likely a typo. The sentence is about biotic stress (pathogens). This has been fixed to biotic.
- Table 6. Institutions are missing for Cuba and Chile. No information? I was not able to find the Institutions and have placed N/A in the table.
- Fig. 1, legend. “One of the many avocado accessions..”. Better to put the number: “One of 60 (correct?) avocado accessions..” This has been added "one of 56...".
- Line 399-400. “Without a reproducible in vitro system, the in vitro cryopreservation protocols will never be successful”. This statement is disputable. What about using cryopreservation without in vitro? USDA developed the system when cryopreserved Citrus buds are grafted directly to greenhouse-grown rootstocks without recovery in vitro. Could this work for avocado? Consider removing this sentence. This sentence was removed from the manuscript.
- Lines 541-545. From my perspective, the information given here would be better placed before conclusion, when describing cryopreservation methods for avocado. Same for Fig. 2.
- Figure 1 is not mentioned in the text. Please check. This has been added to the appropriate place.
- Please fill in required sections (Authors contributions, Funding, Institutional review board statement, Data availability statement and Conflict of interests). Thanks these sections have been filled out.

Reviewer 2 Report
The authors have positively responded to the raised points and I think the paper is now acceptable for publication. However authors need to still pay more attention to the text and corrects some mistakes such as:
Line 117( Error Line! Reference source not found.).
Line 150 (Error! Reference source not found.),
Line 168 (Error! Reference source not found.),
Line 323 (Error! Reference source not found. and Error! 323 Reference source not found.)
Line 389-393 Another study, showed that dehydration at 60 min with sterile air and 30 min in PVS4 at 0 °C produced normal plant development and 100 % survival was obtained after 30, 45 and 60 days minus LN [
Author Response
The authors thankyou for your suggestions in improving the manuscript. The tables and figures references in text have now been fixed.
Line 117( Error Line! Reference source not found.). Table 2 reference inserted.
Line 150 (Error! Reference source not found.),Table 3 reference inserted
Line 168 (Error! Reference source not found.),Table 4 reference inserted
Line 323 (Error! Reference source not found. and Error! 323 Reference source not found.) Table 6 and Figure 1 reference inserted
Line 389-393 Another study, showed that dehydration at 60 min with sterile air and 30 min in PVS4 at 0 °C produced normal plant development and 100 % survival was obtained after 30, 45 and 60 days minus LN [
Deleted minus LN.
This manuscript is a resubmission of an earlier submission. The following is a list of the peer review reports and author responses from that submission.
Round 1
Reviewer 1 Report
This paper is a well-written, comprehensive review of the state-of-art in cryopreservation of avocado, a tree species with recalcitrant seeds. Genetic resources of such species cannot be conserved in conventional seed banks at -18C, and cryopreservation is a feasible alternative to risky and costly field collections. Besides summarizing the available methods and strategies that are currently used for cryopreservation of avocado, the manuscript presents, for the first time, data on number of avocado accessions in genetic banks all over the world obtained through personal communications with genebank curators. I recommend accepting the paper for publication after minor revision based on comments below.
My first, most general, recommendation is making the paper more focused. The title sets a clear frame for the discussion: “tree species with recalcitrant seeds”. However, in the text authors discuss many crops with very different physiology. For example, Table 1 lists also Solanaceae and Mentha, both having orthodox seeds, and Musa, which cultivated accessions are mostly seedless. Similarly, in 3.4 the authors discuss critical factors in cryopreservation using examples of barley, wheat (Line 368), Solanum spp. (458) or Digitalis cell cultures (line 446). Wouldn’t it be more interesting (and more relevant), to discuss the same topics (the roles of preculture, cold acclimatization, antioxidants, cryoprotective agents, etc.) giving examples of cryopreservation protocols developed for avocado or other tree species?
Minor comments
Line 25. Consider removing “structural integrity”.
Line 38. “In vitro conservation and cryopreservation”. Both are ex situ conservation approaches. Consider changing the sentence to “..both traditional in situ and modern ex situ conservation using biotechnological tools (in vitro conservation and cryopreservation”)…”
Line 41. The first two sentences in this paragraph are about in situ conservation but the reference (4) describes ex situ conservation strategies. Please check.
Lines 41-43. Some experts would argue that ex situ conservation is complementary to in situ, i.e. the opposite to your statement.
Line 45. Did you mean “physical” or “physiological”?
Line 62. “Cryopreservation offers a continual supply of material for long-term in vitro research…” I would argue with that. In fact, in most cases in vitro methods provide material supply for cryopreservation. Usually cryopreservation serves as an ultimate back-up of the accessions, and the material is not withdrawn from cryotanks unless it is absolutely necessary.
Table 1. Selection of crops for this table is a little bit confusing. They are not only trees with recalcitrant seeds but also herbaceous species, vines, species with orthodox seeds. Based on what criteria these crops were chosen? Are those collections of clonally propagated crops conserved in vitro? If yes, please mention this in the table title. Why not include, for example, collection of taro and yam at the Centre for Pacific Crops and Trees (CePaCT) (https://www.ncbi.nlm.nih.gov/pmc/articles/PMC6204562/ ; https://www.spc.int/about-us) or German collection of fruit trees (https://link.springer.com/article/10.1007/s11627-017-9841-6), or collections of cassava at CIAT and IITA international genebanks? USDA has Tropical Plant Genetic Resources and Disease Research Unit which conserves tropical fruit trees with recalcitrant seeds (https://www.ars.usda.gov/pacific-west-area/hilo-hi/daniel-k-inouye-us-pacific-basin-agricultural-research-center/tropical-plant-genetic-resources-and-disease-research/docs/main/). A collection of 150 breadfruit accessions is conserved in Tropical Botanical Garde, USA (https://ntbg.org/about/)
Table 1, The international network for the improvement of banana and plantain. Correction to location. In vitro collection of Musa germplasm is physically located in Belgium, not France..
Lines 86-90. Sentence starting with “Over almost 100” is difficult to read. Consider rewriting.
Line 111. “preculture on osmotic media”. Consider adding: “to reduce water content and induce desiccation tolerance”
Line 2.1. Consider changing the title to “Methods to reduce water content” (remove “cryopreservation”). You are describing cryopreservation methods later.
Table 2 and other tables. It is sometimes not clear which description relates to which method. Consider re-formatting.
Table 4. Maybe add that Droplet-Vitrification resembles Vitrification in all steps with only difference that materials are cryopreserved on foil strips.
Table 4. For some methods you give details of preculture, duration of cryoprotectant treatment or dehydration and for others - only number of accessions without protocol details. Consider unifying the format of presenting information here. If you choose to describe the procedure, why not doing that for all references / species mentioned?
Table 5. Those are accessions held in the field, correct? Please add to the title.
Table 5. “A small collection of 46 avocado accessions”. It is definitely larger than the collection of 8 acc. In Nigeria or any other collection below. Please remove “small”.
Table 6. What is the difference between “classical vitrification” and “cryovial vitrification”?
Line 322. Consider changing to “cultivar-dependant in vitro protocols. Would this be correct? The reference (110) is only focused on in vitro propagation.
Line 324 and further. Again, in this section, the description of critical factors in cryopreservation is very general, giving examples of various crops. As a reader, I would prefer seeing examples of the cryopreservation protocols for avocado with the ideas what kind of preculture, cryoprotectants or acclimation work for them and what not. For example, earlier you mentioned 3 races of avocado that are very different in cold tolerance. Does it affect the cryopreservation protocol? Is it possible to use cold acclimation on most cold-tolerant races? Were antioxidants tested for cryopreservation of avocado? Can avocado response to cryopreservation be discussed in relation to cryopreservation protocols developed for other tropical fruit trees?
Line 492. “the higher the number of accessions, the lower the unit cost”. Interesting information. Please give a reference.
Line 503. It is the first time you mention “world’s first repository”.. is it different from the collections in Table 5?
I recommend that Conclusion is a little bit more specific. Could you summarize what cryopreservation methods work best for avocado, how many accessions are now cryopreserved in the genebanks, using what explants (somatic embryos, shoot tips)? What are the most critical factors for cryopreservation of avocado? Is it possible to draw the clear line between the responses of species of different provenance / cold tolerance? What are future prospects for using cryopreservation for avocado and other tree species with recalcitrant seeds?
Reviewer 2 Report
12 have increased instead of has
also: the starting senctence is not clear to me, please rephrase. I would also not use the word cryopreservationspace, maybe area or field?
Recent developments in the cryopreservation space has increased the trend in germplasm collections established through cryopreserved in vitro material.
35 Food?
37,38 Cryopreservation is also ex situ conservation, so cant use it as opposite
43,44 I would say ex situ is complementary, in particular when we talk about wild endangered species. Not the other way round.
45 this sentence is true for living collections but not for cryopreserved cultures. Terms are used very freely here and can lead to confusion by readers. Explain clearer what are ex situ conservation possibilities and give examples
65 I would not put bacteria on the first place, because there is to my knowledge only one example so far, but for virus there are many.
73
the title only mentions recalcitrant seeds, but in fact most of the examples given are vegetatively propagated plants. The importance of cryopreservation for vegetatively propagated crops should be reflected in the title and in the text.
The outlay of all tables is not very clear to read, would suggest modifying them by inserting more columns
138 often used twice, so which one happens often? cant be used for both.
143 use of plural rather than singular?
Table 4: and then placed in a cryo tube ...
I myself was criticized by some reviewers, that I used the term droplet vitrification method when I placed the alfoil first into a cryo vial before dropping it into the LN. They insisted that for the droplet vitrification method, the explants have to be in direct contact with the LN. I think, it should be valid for both, but be prepared for people to be criticial of your statement.
Table 4
Australia: A small collection of
remove a small collection of
212 Avocado (Persea americana Mill.), a high-value fruit native foundin tropical and sub-tropical regions of America insert Komma here? belongs to the family Lauraceae, genus Persea.
native found?
219 if you use was, then please also state in what year.
220 healty foodalternative?
222 explain in more detail.
226 as someone not familiar with the abiotic threats of avocado, I would like to see some examples of the most important breeding goals.
231 here Asia and America is the home for avocados, in 212 it was only America. Explain.
248 abiotic stress ??? Biotic stress in plants is caused by living organisms
249 In Australia the bushfires in 2019/2020 have seen natural forests, bushlands and genetic diversity lost. Many Perseaspp. grow in these forest areas [120]. It has been estimated that we have already lost around 40 % of the forest cover in developing countries of the world through deforestation.
How do these sentences relate to each other?
Makes it sound like Australia is a developing country and the connection between the bushfires and avocado is also not clear.
260 These rootstocks need to have good attributes, such as, salinityand resistance to diseases, ...
remove comma after such as and did you mean salinity tolerance?
263
An example of wild Persea spp.which can be taken advantage of is Persea steyermarkii, which grows adventitious roots from the maintrunk when it is damaged [125] making it a dominant species in a forest in Chiapas and Mexico [128]. In Spain, a fungus called Rosellina necatrix is problematic, however, seedlings from the germplasm bank of the Fundación Salvador Sanchez Colin have shown a tolerance to this disease [129] with a potential to select accessions to combat the fungus.
connection between the two stories is missing, is it the same species, then please say so clearly.
269 Avocado is an open-pollinated plant producing genetically different progeny of seeds, forcing germplasm to be conserved in field repositories (Table 5) instead of seed banking as the seeds are highly heterozygous and recalcitrant [130-131].
recalcitrant is a reason for field conservation but has nothing to do with being open-pollinated. Need to rephrase.
279 somatic embryogenetic cultures? embryogenic
279 the authors jump straight into somatic embryogenesis, there should be a clear introduction to this topic.
290 In the long introduction on cryopreservation there is no mention of cryotube vitrification and classical vitrification, but it is mentioned now. All techniques that are important for avocado should be explained in the introduction.
299 Successwith two Success with two cryopreservation protocols (cryovial and droplet-vitrification) for the conservation of avocado somatic embryos can be applied...
use of success and can be applied in one sentence is unclear to me.
Also, this is actually a general observation, that is not limited to PVS2 protocols: protocols need to be adopted to the cultivars through optimizing exposure times or temperature. Sentence goes with 297 on more research being needed. Rephrase please.
308 Shoot-tip cryopreservation is a novel approachwhich can be used to conserve ‘true-to-type’ avocado plant tissue. It is clonal and true to the accession being preserved without any heterogeneity.
There is nothing novel about shoot tip cryopreservation and even in avocado the reference 152 quoted late in the text is from 2008. Shoot tip propagation is a way of clonal propagation but I would not go as far to say there is no heterogeneity.
310 another new topic/aspect is introduced, there should be a clear introduction so the readers can follow. At the moment there is room for assumptions.
The work is comprehensively described but not always well organised.
Can I ask you and your big team of co-authors to go over the rest of the document yourself. I am sometimes missing clear statements and the use of terminology is not always correct.
I would also consider changing the abstract to make it less general. At the moment it covers cryopreservation per se and provides general knowledge. I would like to see some information related to avocado.
Reviewer 3 Report
Dear Authors,
Manuscript titled: “Cryopreservation for tree species with recalcitrant seeds: the avocado case” is a review related to the important issue of conservation of avocado o (Persea americana Mill.). This is really interesting topic, as avocado is a species which is difficult both to preserve and in vitro culture. Therefore, I think that, this article could be interested for large scientific community. However, there are major issues about this manuscript. First of all the title is inadequate. In the manuscript, Authors do not refer at all to the new techniques of cryopreservation of recalcitrant seeds, e.g., they do not mention the possibility of isolating seed fragments, i.e., embryonic axes or plumuls, and subjecting them to cryopreservation. Besides a general description of the methods used for cryopreservation, most of their attention is focused on cryopreservation of shoot tips. What is worth emphasizing the shoot tips are isolated from shoots not from the seeds as opposed to the germ axes or plum. What is more, shoot tips have nothing to do with the sensitivity to drying of seeds, and sensitivity to drying of seeds is the main, but not only, decisive factor for classifying seeds of certain species into one of the categories: orthodox(desiccation tolerant), intermediate, recalcitrant(desiccation sensitive).
I agree that shoot tip is an very important organ for cryopreservation of plant tissues, especially if we want to preserve a specific genotype. However, it is completely incomprehensible to me that the authors have attempted to demonstrate the advantage of cryopreservation of the shoot tip over cryopreservation of seeds, especially in case of ex situ conservation. This is demonstrated by statements such as:
“seed conservation does not serve the purpose of germplasm preservation”
“Alternatively, as seeds do not carry the same genetic make-up as the mother plant, especially in the context of woody rainforest species of which the cross-pollination is dominant; seed conservation does not serve the purpose of germplasm preservation’
These types of statements testify to the fact that the manuscript was written in a hurry and some generalisations were used. It should be noted that the preservation of gene resources, both in situ and ex situ, is aimed at preserving the greatest possible biodiversity of the material preserved. Therefore, the seeds are ideal for this purpose, and shoot tips are not a good choice, because it requires the preservation of many shoot tips (at least several hundred individuals), which is an extremely laborious and expensive process. On the other hand, shoot tips are ideal for preserving selected genotypes, for this reason seeds are completely unsuitable due to the fact that each seed is a different individual.
Also, the title of chapter 2 is incomprehensible to me, because in this chapter the authors describe all the methods not related to recalcitrant seeds, which in a way duplicates the same error as with the title.
Moreover, some states from Chapter 2 on cryopreservation of shoot tips are almost repeated in Chapter 4, making the work, in my opinion, too long and difficult to follow.
Also such statement as: “…are sensitive to desiccation, chilling and freezing stress, making them unsuitable for seed banking or cryopreservation”
Testify that the authors did not take the trouble to read the literature on cryopreservation of seeds, especially the area related to cryopreservation isolated from part of seeds or embryos form recalcitrant seed.. The authors, as cryopreservators, should know that the fact that at this point in time there is no described procedure for cryopreservation of seeds or fragments of seeds does not make this impossible and that it is not possible on this species.
Therefore, in my opinion, the manuscript in its current form is not suitable for publication in Plants.